# Vitamin D Status among Women in a Rural District of Nepal: Determinants and Association with Metabolic Profile—A Population-Based Study

**DOI:** 10.3390/nu14112309

**Published:** 2022-05-31

**Authors:** Chandra Yogal, Marianne Borgen, Sunila Shakya, Biraj Karmarcharya, Rajendra Koju, Mats P. Mosti, Miriam K. Gustafsson, Bjørn Olav Åsvold, Berit Schei, Astrid Kamilla Stunes, Unni Syversen

**Affiliations:** 1Department of Clinical and Molecular Medicine, Faculty of Medicine and Health Science, Norwegian University of Science and Technology, 7491 Trondheim, Norway; chandra.m.yogal@ntnu.no (C.Y.); marborg@stud.ntnu.no (M.B.); mats.p.mosti@ntnu.no (M.P.M.); miriam.gustafsson@ntnu.no (M.K.G.); kamilla.stunes@ntnu.no (A.K.S.); 2Department of Community Program, Kathmandu University School of Medical Science, Dhulikhel Hospital, Kathmandu University Hospital, Dhulikhel 45200, Nepal; birajmk@gmail.com; 3Department of Gynecology and Obstetrics, Kathmandu University School of Medical Sciences, Dhulikhel Hospital, Kathmandu University Hospital, Dhulikhel 45200, Nepal; sunilashakya@yahoo.com; 4Department of Internal Medicine, Dhulikhel Hospital, Kathmandu University Hospital, Dhulikhel 45200, Nepal; rajendrakoju@gmail.com; 5Regional Education Center, Helse Midt-Norge, 7030 Trondheim, Norway; 6K.G. Jebsen Center for Genetic Epidemiology, Department of Public Health and Nursing, Faculty of Medicine and Health Science, Norwegian University of Science and Technology, 7491 Trondheim, Norway; bjorn.o.asvold@ntnu.no; 7Department of Endocrinology, Clinic of Medicine, St. Olavs University Hospital, 7030 Trondheim, Norway; 8Department of Public Health and Nursing, Faculty of Medicine and Health Science, Norwegian University of Science and Technology, 7491 Trondheim, Norway; berit.schei@ntnu.no

**Keywords:** vitamin D deficiency, metabolic profile, milk intake, women, rural Nepal

## Abstract

Hypovitaminosis D is prevalent worldwide, and especially in South-Asia. According to the Institute of Medicine (IOM), 25(OH)D levels below 30 nmol/L are defined as vitamin D deficiency (VDD) and levels between 30–50 nmol/L as insufficiency (VDI). Besides its role in calcium homeostasis, it has been postulated that vitamin D is involved in metabolic syndrome. Given the scarcity of data on vitamin D status in Nepal, we aimed to examine the prevalence of VDD and VDI, as well as the determinants and association with metabolic parameters (lipids, HbA1c), in a cohort of women in rural Nepal. Altogether, 733 women 48.5 ± 11.7 years of age were included. VDD and VDI were observed in 6.3 and 42.4% of the participants, respectively, and the prevalence increased by age. Women reporting intake of milk and eggs > 2 times weekly had higher 25(OH)D levels than those reporting intake < 2 times weekly. Women with vitamin D levels < 50 nmol/L displayed higher levels of cholesterol, LDL-cholesterol, triglycerides, and HbA1c. Additionally, a regression analysis showed a significant association between hypovitaminosis D, dyslipidemia, and HbA1c elevation. In conclusion, VDI was prevalent and increased with age. Milk and egg intake > 2 times weekly seemed to decrease the risk of VDI. Moreover, hypovitaminosis D was associated with an adverse metabolic profile.

## 1. Introduction

Vitamin D is obtained through the diet or by synthesis in the skin after exposure to UVB radiation from the sun [1,2]. However, dietary sources of vitamin D are scarce and sun exposure is often limited [3,4]. Consequently, hypovitaminosis D is prevalent and has become a major global health burden, affecting a billion people worldwide [5,6,7]. Serum levels of 25-hydroxy-vitamin D (25(OH)D) are used to evaluate vitamin D status. There has been some controversy regarding optimal serum 25(OH)D concentrations and definitions of vitamin D deficiency (VDD), insufficiency (VDI), and sufficiency [8]. According to the Institute of Medicine (IOM), 25(OH)D levels below 30 nmol/L are defined as vitamin D deficiency (VDD) and levels between 30–50 nmol/L as insufficiency (VDI).

The prevalence of hypovitaminosis D seems to be high, especially among South Asians, despite their living in latitudes that enable dermal synthesis of vitamin D [9,10]. A systematic review and meta-analysis published 2021, including 65 studies from five South Asian countries and 44,717 participants, reported a pooled prevalence of VDI (<50 nmol/L) of 68%. The highest prevalence was found in Pakistan (73%), followed by Bangladesh (67%), India (67%), Nepal (57%), and Sri Lanka (48%) [10]. However, the data from Nepal were based on only two studies. One study was hospital-based and included 300 patients of both sexes, while the other included 500 lactating women from a semi-urban area [11,12]. Studies addressing vitamin D status in the general population are lacking.

Vitamin D plays a pivotal role in facilitating intestinal calcium absorption and is also involved in bone mineralization [13,14]. Severe deficiency causes rickets in children and osteomalacia in adults [15]. Besides the classical functions, vitamin D is implicated in metabolic syndrome, obesity, diabetes, respiratory diseases, cancer, and cardiovascular disease [3,16,17,18,19]. Moreover, hypovitaminosis D during pregnancy has been associated with increased risk of metabolic syndrome and osteoporosis in the offspring [20,21]. This is of concern, as several studies from low- and middle-income countries have reported lower vitamin D levels in females than in their male counterparts [3].

There is a knowledge gap concerning vitamin D status in the general population in Nepal, especially in rural settings. Given that women are more prone than men to hypovitaminosis D, we aimed to assess the prevalence and determinants of VDD and VDI and the association with metabolic parameters among a female population in a rural district of Nepal.

## 2. Materials and Methods

### 2.1. Study Design, Study Site and Participants

The present study is part of a large cross-sectional survey conducted during October–December 2019 among women in a rural setting of Nepal. The main objective was to determine the prevalence of diabetes, risk factors, complications, and relation with vitamin A and D. Participants were recruited from a cohort of 1498 married, non-pregnant women > 21 years who participated in a study in 2012–2013 addressing reproductive health [22,23] and non-communicable diseases (NCDs). As in the previous study, only non-pregnant women were included, and they were excluded in case of physical and mental conditions that made it challenging to participate. The study site, i.e., the Kavre District, is located at a latitude of about 27.5° N, with an average elevation of 1890 m above sea level.

### 2.2. Recruitment of Study Subjects

Female community health volunteers (FCHVs) appointed by the government to run preventive health programs were involved in the recruitment. They were informed about the purpose of the study and the procedures. Thereafter, FCHV visited the women who participated in the abovementioned study. They were informed about the new study and invited to participate. One-day screening sites were prepared at health centers, local schools, and village halls. In the morning on the day of screening, the women were informed by PhD student Chandra Yogal about the purpose and procedures, and were told that they could withdraw from the study at any time without consequence. Informed consent in the form of a signature or thumb print was obtained from women who agreed to participate.

### 2.3. Data Collection

Open data kit (ODK) free software (aggregate V1.4.11, University of Washington, Seattle, WA, USA) was used to collect data. Data were collected in three steps: an interview, through physical measurements, and blood sample collection. A comprehensive questionnaire administered by four trained health workers was completed (Appendix A). Data on sociodemographic parameters, tobacco and alcohol use, dietary factors, and use of calcium and vitamin D supplements were collected. Dietary factors included consumption of rice, vegetables, instant noodles, eggs, milk and other dairy products. Participants were asked for frequency of intake but not amount. Body weight was measured by a portable digital weighing scale (secca220, Hamburg, Germany). Participants were requested to stand barefoot on the digital weighing scale; measurements were made in kilograms. Height measurements (cm) were performed with a stadiometer attached on the wall surface. Body mass index (BMI, kg/m^2^) was calculated. Waist circumference (WC) was measured in cm in standing position with a non-stretchable measuring tape. The measurement was made at the end of a natural expiration, at the midpoint between the lower margin of the last palpable rib in the midaxillary line and the top of the iliac crest. Blood pressure (BP) in the left arm was measured twice in a sitting position by a digital device (Omron-5 series digital blood pressure monitor), after 15–30 min of the interview, and the end of the interview. The average reading was taken for the analysis.

### 2.4. Blood Sample Collection and Analyses

Fasting blood samples were collected from a cubital vein. Whole blood was kept on ice and transported the same day to the Department of Biochemistry, Dhulikhel hospital (DH), Kathmandu University hospital for analysis of glycosylated hemoglobin (HbA1c) by Hb-Vario-For HbA1c test, based on High Performance Liquid Chromatography (HPLC) by Erba Diagnostic Mannheim GmbH, Germany. The remaining blood was centrifuged. Serum samples were stored for 3–4 h at −2 to 8 °C and further at −30 °C degrees at the study site before being transported to DH in cold boxes and stored at −80 °C until analyses. Analyses of 25(OH)D, calcium, phosphate, and parathyroid hormone (PTH) were performed with LIAISON^®^ analyzer with chemiluminescence immunoassay (CLIA) technology by DiaSorian, Italy, at DH. Serum levels of total cholesterol, LDL-cholesterol (LDL-C), HDL-cholesterol (HDL-C), and triglycerides (TG) were measured by the enzymatic spectrophotometric method using a BA 400 full automatic analyzer, BioSystems S.A. Spain. Reference values are presented in Appendix A. Internal quality control was performed in the laboratory every day for the 25(OH)D, PTH, lipid profile, calcium, and phosphate analyses by Accusera Internal quality control material, manufactured by Randox laboratories, UK. The policy is to accept plus/minus 2 standards for quality control and reject if it is plus/minus 3 or more. External quality assurance was done for calcium, phosphate, and lipid profile under Christian Medical College Vellore. There was no external quality assurance service for vitamin D and PTH.

### 2.5. Outcome Variables and Criteria Used to Define Vitamin D status

The primary outcome of this study was vitamin D status measured as total serum 25(OH)D. We used the recommendations of IOM for classification of vitamin D status, whereby < 30 nmol/L (12 ng/mL) represents deficiency and 25(OH)D 30–50 nmol/L (20 ng/mL) an insufficiency [8]. However, since the number of individuals with VDD was low in our population, we chose to compare those with levels below and above 50 nmol/L. All blood samples were collected in the same season (October–December), and adjustment for season was therefore not performed.

### 2.6. Statistics

Continuous variables are presented as mean with standard deviation, (SD) and categorical variables as counts and percentage. Group differences were analyzed using one-way ANOVA test with post hoc Dunnett test for comparison with continuous variables and Pearson’s χ^2^ (or Chi square) test for categorical variables. Pearson’s correlation coefficient was used to estimate the correlation between 25(OH)D concentrations and different parameters. Binary logistic regression analysis was performed to examine the associations between outcome variable and covariates and presented as crude odds ratios (COR) and adjusted odds ratio (AOR) (adjusted for age and BMI) with 95% confidence intervals (CIs). All analyses were performed using the IBM SPSS statistics (version 28.0.1.0 (142), New York, NY, USA). A *p*-value < 0.05 was considered as significant.

### 2.7. Ethical Considerations

This study was approved from the National Health Research Council, Nepal (Reg.no. 744/2018), the Institutional Review Committee of Kathmandu University School of Medical Sciences/Dhulikhel Hospital (approval no. 124/19) and Regional Committees for Medical and Health Research Ethics, Norway (REK no. 13003). Informed written consent was obtained from the participants. The study was conducted according to the guidelines provided in the Declaration of Helsinki [24].

## 3. Results

### 3.1. Characteristics of the Study Population

A total of 813 women were enrolled in the study; of these, 29 were excluded as they declined to give blood or were not able to provide a sufficient quantity. Moreover, blood samples were collected from 51 subjects, but no other data were obtained. Finally, 733 women, mean age 48.5 ± 11.7 years who completed all three steps of data collection (interview, blood collection, physical measurements) were included in the analyses. Table 1 shows sociodemographic and lifestyle factors and corresponding mean 25(OH)D levels. Mean 25(OH)D level of the study population was 51.6 ± 16.0 nmol/L. A significant decline in 25(OH)D level occurred with increasing age. Most of the study population (83.4%) belonged to the Adhivasi/Janajati ethnicity that may be referred to as the middle class, followed by Brahmin/Chhetri (12.0%) and Dalit (4.6%) ethnicity, the high and low casts, respectively. The Dalit ethnicity displayed lower 25(OH)D level than the other ethnicities. A total of 83.6% were uneducated, and they were found to have a lower mean level of 25(OH)D than educated. Agriculture was the major source of income (77.5%). The mean number of children was 3.5 ± 1.6 per women. Those who had never given birth (19/733) exhibited lower mean 25(OH)D levels than those with 1–3 and 3–10 child births. Current smoking was reported by 16%; these subjects had lower 25(OH)D levels than never and former smokers. Daily alcohol intake was reported by 17%. Rice intake > 2 times daily was reported by the majority. Women with milk intake ≥ 2 times per week had significantly higher mean 25(OH)D concentration than those with intake < 2 times a week. No difference in 25(OH)D levels was observed between those reporting intake of other dairy products > or < than two times a week. Mean 25(OH)D level was significantly higher in subjects with egg consumption >2 times weekly versus <2 times. Intake of vitamin D and calcium supplementation was reported by 36 and 43 women, respectively. We do not have information concerning dosage or adherence. Only 8.3% and 7.0% of these had vitamin D levels > 75 nmol/L. Mean 25(OH)D concentrations did not differ between women taking or not taking vitamin D supplementation.

### 3.2. Prevalence of VDD and VDI

Figure 1 shows the vitamin D status in the population. Altogether, 6.3% displayed VDD, and 42.4% were insufficient in vitamin D. No substantial change in prevalence was observed when adding the 51 women who were excluded from the study due to lack of data except for vitamin D measurement. Only five subjects had serum 25(OH)D levels more than 100 nmol/L. Eight of the women with VDD had elevated PTH levels. PTH elevation was seen in 67 women, and among them, 31 subjects had vitamin D level 30–50 nmol/L and 24 had levels between 50–75 nmol/L, consistent with secondary hyperparathyroidism. Eighteen women had hypocalcemia with an even distribution between categories of vitamin D levels, only four of these had increased PTH level.

Table 2 shows prevalence of hypovitaminosis by sociodemographic and lifestyle factors. A high prevalence was observed among uneducated, women with no children, those with intake of eggs and milk intake (ns) < 2 times weekly, former and current smokers (ns) and women with current alcohol intake (ns).

Table 3 shows anthropometric measurements, blood pressure, and biochemical parameters stratified by vitamin D status. Mean weight, height, BMI, and WC were 54.6 ± 10.2 kg, 148.5 ± 6.6 cm, 24.8 ± 4.8 kg/m^2^, and 78.1 ± 11.0 cm, respectively. Mean HbA1c was 37.4 ± 8.4 mmol/mol, mean TG 1.4 ± 1.0 mmol/L, and mean HDL-C 1.3 ± 0.4 mmol/L. Women with hypovitaminosis D were older, displayed a lower body weight, and lower BMI and WC, although the latter two variables were not significant. Moreover, levels of PTH, phosphate, HbA1c, total cholesterol, LDL-C, and TG were higher compared to women with vitamin D levels > 50 nmol/L.

### 3.3. Factors Associated with Hypovitaminosis D

Table 4 shows the correlation between serum 25(OH)D, age, anthropometrics, BP, and biochemical parameters. A weak but significant inverse correlation was observed for most of the parameters. Only the anthropometric parameters correlated positively.

A binary logistic regression analysis was performed to measure the association between hypovitaminosis D (25(OH)D < 50 nmol/L) and sociodemographic characteristics, lifestyle factors, anthropometrics, BP, and metabolic parameters (Table 5). Adjustments were performed for age and BMI. The odds of hypovitaminosis D increased with age, with women 60–80 years displaying the highest odds, i.e., COR 5.7 (95% CI: 2.56–12.49, *p* > 0.001), compared to the 21–30 age group. Women with no children exhibited increased risk, whereas those with 3–10 children were less likely (AOR 0.7, 95% CI: 0.51–1.02, *p* = 0.066) to be vitamin D deficient or insufficient compared to women with 1–3 children. Milk intake ≥ 2 times weekly seemed to reduce the risk of hypovitaminosis D. High levels of HbA1c, total cholesterol, LDL-C, and TG were associated with VDI.

## 4. Discussion

In this large, comprehensive study among a female population in a rural district of Nepal, we observed a high prevalence of hypovitaminosis D, i.e., 6.3% displaying VDD and 42.4% VDI. The study subjects were mainly uneducated with agriculture as the primary occupation. The prevalence of VDD and VDI increased by age. Women reporting intake of milk and eggs > 2 times weekly had higher vitamin D levels compared to those with intake < 2 times a week. Moreover, women with vitamin D levels below 50 nmol/L displayed higher mean levels of HbA1, total cholesterol, LDL-C and TG. These findings were confirmed by regression analysis showing significant associations between low levels of vitamin D and HbA1c elevation, as well as dyslipidemia. 

In the present study, almost half of the participants had vitamin D levels below 50 nmol/L, whereas 46 of the women (6.3%) had levels below 30 nmol/L. In concordance with previous studies [25], we found that aging was the strongest risk factor for VDD and VDI. This could be attributed to less out-door activities and to attenuation of the dermal production of vitamin D with age. There are few studies for comparison, as previous studies addressing vitamin status in Nepal mainly were conducted in pregnant or lactating women [12], in children [26], or in hospitalized subjects [27]. Haugen et al. demonstrated a high prevalence of hypovitaminosis D among 500 lactating women from a semi-urban area Bhaktapur, 59.8% displaying 25(OH)D levels < 50 nmol/L and 14% levels < 30 nmol/L. Despite this, their infants had adequate vitamin D status [12]. This finding was assumed to be attributed to the Nepalese tradition of outside breastfeeding, exposing newborn infants to sunlight [26]. In contrast, Avagyan et al. observed hypovitaminosis D (25(OH)D < 50 nmol/L) in 91.1% of 270 children aged 12–60 months, selected randomly from the records of a vitamin A supplementation program [26]. About one third of the children had s-25(OH)D concentrations below 25.1 nmol/L, among them 4% had severe VDD (<12.5 nmol/L). Whether the children had skeletal affection was not mentioned. Interestingly, children who were currently breast-fed displayed higher 25(OH)D levels than those who were not.

A cross-sectional study, conducted at Grande International Hospital, Kathmandu, from January to December 2019 included 7075 patients, 2289 males and 4786 females. Female patients displayed a higher prevalence of VDD than male patients, 20.49% versus 14.02%, respectively, whereas the rate of VDI was similar [28]. In a retrospective study including 3320 subjects who were tested for VDD, 84.5% displayed levels < 30 ng/mL (75 nmol/L) and 25.9% had levels < 10 ng/mL (25 nmol/L) [27]. Bhatta et al. examined vitamin D status among 2158 individuals 19–60 years in the western region of Nepal, and reported that 73.68% had vitamin D levels < 25 ng/mL (<62.4 nmol/L) [29]. As in most studies, women were more susceptible to hypovitaminosis D.

The prevalence of VDD of 6.3% in the present study was somewhat lower than that reported by Cashman et al. among 55,844 Europeans; that study reported that 13% displayed levels below 30 nmol/L [30]. This was a pooled estimate, including all age groups, ethnicities, and latitudes. Seasonal differences were observed with a prevalence of 17.7% during winter (October–March) and 8.3% in the summer (April–November). The prevalence of VDD was 3 to 71-fold higher among dark-skinned ethnic subgroups compared to the white population. In the total population, 40.4% had vitamin D levels below 50 nmol/L compared to 48.7% in the present study [30].

The high prevalence of hypovitaminosis D observed in the present study reflects the low content of vitamin D in the diet, as well as a lack of sun exposure. Notably, Nepal has more than 300 sunny days a year, and Nepalese do not avoid the sun for cultural reasons. The location of the study site is ideal with respect to altitude and latitude. Moreover, most of our study participants received their main income from agriculture, and accordingly, had to spend a substantial amount of time in the fields. Factors that could prevent sun exposure include pigmentation of the skin, clothing covering major parts of the body, and use of sunscreen [3,31]. We do not have data on use of sunscreen, but it is apparent that dark skin color and clothing habits of our study subjects may reduce the impact of UVB radiation on dermal synthesis of vitamin D. Clothing appears to be similar throughout the year with respect to the area covered. Other factors that could affect the UVB radiation are season, altitude, and pollution. In the present study, serum samples were collected during October-December, which is the season with the least rainfall. The lack of clouds contributes to a more pronounced UVB radiation. This is supported by the study of Bhatta et al. in western Nepal, showing seasonal differences with the highest vitamin D levels in the autumn [29]. The location of our study site at 1890 m above sea level will also result in a higher dosage of UVB radiation. On the other hand, air pollution tends to reduce the radiation. Noteworthy, Kathmandu is one of the cities with the highest air pollution. The air in the Kathmandu Valley contains ten times the pollutant concentration set forth by WHO guidelines [32,33].

Most diets contain only scarce amounts of vitamin D; as such, fortification or supplements may be necessary. In the present study, vitamin D supplements were used by very few individuals, and serum levels of vitamin D were not affected. Rice and vegetables are the main ingredients in the diet of this rural population. Because of increasing urbanization, fast food, like instant noodles, has also been incorporated in the diet. None of these foods contain vitamin D. To the best of our knowledge, no foods in Nepal are fortified with vitamin D. Access to fish is very low in the region where the study was conducted. Eggs are a known source of vitamin D, and subjects reporting egg consumption > 2 times weekly exhibited a higher vitamin D concentration compared to <2 times a week. About 50% reported milk intake more than two times weekly. Notably, those who reported intake of milk more than 2 times weekly displayed a higher mean level of 25(OH)D, and VDD/VDI was less common. This was confirmed by logistic regression analysis showing reduced risk for hypovitaminosis D among women with milk intake > 2 times weekly, although borderline significant. Milk is not fortified with vitamin D in Nepal. However, cows are capable of vitamin D synthesis. This was explored in a Danish study, which demonstrated that vitamin D synthesis in dairy cattle takes place in all areas of the skin and not exclusively in skin areas where hair coverage is scant or lacking [34]. Moreover, another Danish study demonstrated that cows exposed to artificial UVB radiation experienced a rise in vitamin D levels in plasma and milk to levels similar to those of cows grazing at pasture during the summer [35]. They proposed that artificial UVB radiation could be applied to ensure a high vitamin D level in cows raised all year indoors. Given that the cows in Nepal are mainly kept out-door, it is reasonable that they have access to sufficient UVB radiation from the sun for dermal synthesis of vitamin D all year through. Whether this is reflected in a high vitamin D concentration in their milk remains to be verified. Studies on lactating women indicate that the concentration of vitamin D in human milk correlates with their intake of vitamin D and UVB exposure [36].

Hypovitaminosis D has been associated with metabolic syndrome, diabetes, and cardiovascular disease [37,38]. In the present study, we explored the relationship between vitamin D levels < 50 nmol/L and a range of metabolic parameters. We observed lower BMI and WC among those with VDI. This concords with the study by Haugen et al., which showed a positive association between BMI and 25(OH)D levels among lactating mothers [12]. Our findings are, however, in contrast to those of previous studies reporting an inverse association between BMI and 25(OH)D [39,40]. The higher levels of HbA1c, total cholesterol, LDL-C, and TG among our study subjects with hypovitaminosis D indicate that vitamin D may be involved in the pathogenesis of metabolic dysfunction. This was supported by a regression analysis showing a significant association between hypovitaminosis D and elevated levels of HbA1c and dyslipidemia. Our findings are in concordance with those of previous studies showing a relationship between VDD and metabolic syndrome in women [16,19,41]. Our study is the first to demonstrate this association among women in rural Nepal.

The high prevalence of hypovitaminosis D among women in rural Nepal is of concern. It is well known that low vitamin D levels have negative skeletal effects, as exemplified in our population, by inducing secondary hyperparathyroidism and thereby stimulating bone resorption. Secondary hyperparathyroidism was also observed among 24 women with vitamin D levels between 50 and 75 nmol/L. The association of hypovitaminosis D and an adverse metabolic profile makes it reasonable to assume that VDD is contributing to the metabolic syndrome and diabetes epidemics. Low maternal vitamin D levels during pregnancy have also been associated with future disease in the offspring, e.g., both osteoporosis and metabolic dysfunction [20,21]. To ensure sufficient vitamin D levels, it is necessary to focus on vitamin D supplements and fortification in food. Based on our findings, eggs, milk, and other dairy products are sources of vitamin D, and intake of these should be encouraged [34]. The enhancement of serum levels of vitamin D in the population may ultimately contribute to improved public health.

The strengths of this study are its large sample of rural women and the acceptable participation rate (56.3%), as well as the number of specimens obtained. There were, however, several limitations. We did not have data on time spent in the sun, covering with clothes, or use of sunscreen. Data on the intake of different food items were based on reported frequency but not amount. The vitamin D contents of foods and supplements were not determined. The assay used for analyses of serum 25(OH)D in the present study (CLIA) is not the gold standard, and the prevalence of VDD and VDI may have been overestimated. Moreover, the laboratory at DH does not take part in the international standardization program, and instead, internal quality control is performed. The study was limited to women in a rural, hilly district of Nepal, and may not translate to women in the flatlands or in urban districts, or to men.

## 5. Conclusions

In conclusion, we observed a high prevalence of hypovitaminosis D among married women in a rural district of Nepal. Age was an important predictor of vitamin D status. Milk and eggs seemed to be major sources of vitamin D. Moreover, VDD was associated with high HbA1c and dyslipidemia, implying a possible role of VDD in the pathogenesis of metabolic syndrome.

## Figures and Tables

**Figure 1 nutrients-14-02309-f001:**
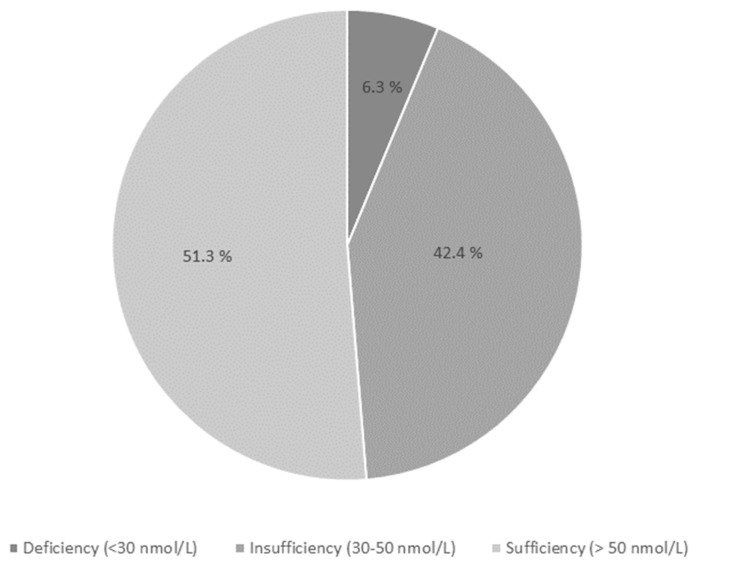
Vitamin D status in the study population.

**Table 1 nutrients-14-02309-t001:** Sociodemographic and lifestyle factors by serum 25(OH)D concentration of the study population.

Characteristics (*n* = 733)	*n* (%)	Serum 25(OH)D Mean ± SD	*p* Value
Age, (years),		48.5 ± 11.7	
Serum 25(OH)D, nmol/L		51.6 ± 16.0	
Age groups (years)			<0.001
21–30	43 (6.0)	60.3 ± 13.5	*
31–40	164 (22.0)	53.6 ± 13.8	0.037
41–50	227 (31.0)	51.8 ± 15.0	0.004
51–60	190 (26.0)	50.1 ± 15.6	<0.001
61–80	109 (15.0)	47.3 ± 20.6	<0.001
Ethnicity			0.047
Brahmin/Chhetri	89 (12.0)	50.3 ± 20.1	*
Adhivasi/Janajati	611 (83.4)	52.1 ± 15.3	0.499
Dalit	33 (4.6)	45.5 ± 14.8	0.236
Educational status ^a^			0.036
Uneducated	613 (83.6)	51.1 ± 16.3	
Educated	108 (16.4)	54.6 ± 14.0	
Number of children			0.008
Null	19 (2.6)	41.1 ± 11.3	*
1–3	377 (51.4)	52.4 ± 14.3	0.009
3–10	337 (46.0)	51.3 ± 17.9	0.019
Milk intake ^b^			0.031
≥2 times a week	273 (49.0)	52.8 ± 16.0	
<2 times a week	282 (51.0)	50.1 ± 14.0	
Egg consumption ^c^			<0.001
≥2 times a week	395 (81.6)	52.9 ± 16.0	
<2 times a week	89 (18.4)	35.6 ± 4.9	
Instant noodle intake ^d^			0.268
≥2 times a week	226 (37.4)	52.7 ± 15.0	
<2 times a week	379 (62.6)	51.3 ± 15.1	
Smoking			0.006
Never	478 (65.2)	53.0 ± 16.2	*
Former	115 (19.1)	49.6 ± 15.0	0.015
Current	140 (15.7)	48.9 ± 15.6	0.038
Alcohol intake			0.816
Never	511 (70.0)	51.5 ± 15.8	
Former	96 (13.0)	51.3 ± 17.1	
Current	126 (17.0)	52.6 ± 16.0	

Abbreviations: SD, standard deviation. Continuous data are presented as mean values with standard deviations; categorical data are presented as numbers and percentages in parentheses. * Reference group for post hoc, Dunnett test for comparison; ^a^ missing *n* = 12; ^b^ missing *n* = 178; ^c^ missing *n* = 249; ^d^ missing *n* = 128.

**Table 2 nutrients-14-02309-t002:** Prevalence of hypovitaminosis D by sociodemographic and lifestyle factors.

Characteristics	Serum 25 (OH)D Level		*p* Value
Study population (*n* = 733), *n* (%)	(<50 nmol/L)	(>50 nmol/L)	
Age groups, (years)			<0.001
21–30	11 (25.6)	32 (74.4)	
31–40	63 (38.4)	101 (61.6)	
41–50	109 (48.0)	118 (52.0)	
51–60	102 (53.7)	88 (46.3)	
61–80	72 (66.0)	37 (34.0)	
Ethnicity			0.055
Brahmin/Chhetri	50 (56.2)	39 (43.8)	
Adhivasi/Janajati	286 (46.8)	325 (53.2)	
Dalit	21 (63.6)	12 (36.4)	
Educational status ^a^			0.046
Uneducated	308 (50.2)	305 (49.8)	
Educated	43 (40.0)	65 (60.2)	
Number of children			0.050
Null	14 (73.7)	5 (26.3)	
1–3	174 (46.2)	203 (53.8)	
>3	169 (50.1)	168 (49.9)	
Milk intake ^b^			0.065
≥ 2 times a week	119 (43.6)	154 (56.4)	
< 2 times a week	145 (51.4)	137 (48.6)	
Egg intake ^c^			<0.001
≥ 2 times a week	182 (46.0)	213 (54.0)	
<2 times a week	89 (100.0)	0 (0.0)	
Instant noodle intake ^d^			0.316
≥2 times a week	102 (45.0)	124 (55.0)	
<2 times a week	187 (49.3)	192 (50.7)	
Smoking			0.071
Never	218 (45.6)	260 (54.4)	
Former	63 (54.8)	52 (45.2)	
Current	76 (54.3)	64 (45.7)	
Alcohol intake			0.303
Never	249 (48.7)	262 (51.3)	
Former	41 (42.7)	55 (57.3)	
Current	67 (53.2)	59 (46.8)	

^a^ missing *n* = 12; ^b^ missing *n* = 178; ^c^ missing *n* = 249; ^d^ missing *n* = 128.

**Table 3 nutrients-14-02309-t003:** Anthropometrics, blood pressure and biochemical parameters stratified by vitamin D status.

Characteristics	Total	Insufficiency (<50 nmol/L)	Sufficiency (≥50 nmol/L)	*p* Value
Total population, *n* (%)	733	357 (48.7)	376 (51.3)	
Age, years	48.5 ± 11.7	51.0 ± 11.7	46.1 ± 11.2	<0.001
Height, cm	148 ± 6.7	148.0 ± 6.5	149.0 ± 6.8	<0.046
Weight, kg	54.7 ± 10.2	53.5 ± 10.0	55.8 ±10.4	0.003
BMI ^a^, kg/m^2^	24.8 ± 4.8	24.4 ± 4.2	25.2 ± 5.2	0.027
WC ^a^, cm	78.0 ± 11.0	77.1 ± 10.4	78.9 ± 11.3	0.021
SBP, mmHg	125.5 ± 19.0	127.1 ± 21.1	124.0 ± 16.8	0.029
DBP, mmHg	81.2 ± 10.8	82.1 ± 11.5	80.4 ± 10.1	0.032
Serum 25(OH)D, nmol/L	51.6 ± 16.0	48.9 ± 7.4	63.7 ± 12.2	
PTH, pg/ml	24.2 ± 10.6	25.3 ± 9.2	23.2 ± 11.8	0.007
Calcium ^b^, mmol/L	2.3 ± 0.1	2.3 ± 0.1	2.3 ± 0.1	0.753
Phosphate ^c^, mmol/L	1.2 ± 0.2	1.3 ± 0.1	1.2 ± 0.2	<0.001
HbA1c ^d^, mmol/mol	37.4 ± 8.4	38.3 ± 10.1	36.5 ± 6.3	0.003
Total cholesterol c, mmol/L	4.6 ± 1.0	4.8 ± 1.0	4.4 ± 1.0	<0.001
LDL-C ^e^, mmol/L	2.6 ± 0.8	2.8 ± 0.8	2.5 ± 0.8	<0.001
HDL-C ^c^, mmol/L	1.3 ± 0.4	1.3 ± 0.4	1.2 ± 0.3	0.022
TG ^c^, mmol/L	1.4 ± 1.0	1.7 ± 0.8	1.2 ± 0.6	<0.001

Abbreviations: SD, standard deviation; BMI, body mass index; 25(OH)D, 25-hydroxyvitamin D; PTH, parathyroid hormone, HbA1c, glycated hemoglobin; LDL-C, low density lipoprotein choles-terol; HDL-C, high-density lipoprotein cholesterol; TG, triglycerides; ^a^ missing *n* = 3; ^b^ missing *n* = 7; ^c^ missing *n* = 6; ^d^ missing *n* = 10; ^e^ missing *n* = 8.

**Table 4 nutrients-14-02309-t004:** Correlation between serum 25(OH)D and anthropometric measurements, blood pressure and biochemical parameters in the study population.

Characteristics	Coefficient (r)	*p* Value	*n*
Age, years	−0.180	0.001	733
Height, cm	0.064	0.082	732
Weight, kg	0.112	0.002	731
BMI, kg/m^2^	0.083	0.025	730
Waist circumference, cm	0.080	0.031	730
Systolic BP, mmHg	−0.034	0.361	733
Diastolic BP, mmHg	−0.037	0.313	733
PTH, pg/ml	−0.128	<0.001	733
Calcium, mmol/L	−0.038	0.312	726
Phosphate, mmol/L	−0.150	<0.001	727
HbA1c, mmol/mol	−0.080	0.032	723
Total cholesterol, mmol/L	−0.213	<0.001	727
LDL-C, mmol/L	−0.122	<0.001	725
HDL-C, mmol/L	−0.115	0.002	727
TG, mmol/L	−0.214	<0.001	727

Abbreviations: BMI, body mass index; PTH, parathyroid hormone; HbA1c, glycated hemoglobin; LDL-C, low density lipoprotein cholesterol; HDL-C, high-density lipoprotein cholesterol; TG, triglycerides.

**Table 5 nutrients-14-02309-t005:** Factors associated with hypovitaminosis D in the study population.

Characteristics (*n* = 733)	COR (95% CI)	*p* Value	AOR (95% CI)	*p* Value
Age groups (years)				
21–30 (*n* = 43)	1			
31–40 (*n* = 164)	1.8 (0.85–3.85)	0.121		
41–50 (*n* = 227)	2.7 (1.29–5.59)	0.008		
51–60 (*n* = 190)	3.4 (1.61–7.08)	0.001		
61–80 (*n* = 109)	5.7 (2.56–12.49)	<0.001		
Ethnicity				
Adhivasi/Janajati (*n* = 611)	1		1	
Brahmin/Chhetri (*n* = 89)	1.5 (0.93–2.28)	0.100	1.5 (0.94–2.36)	0.093
Dalit (*n* = 33)	2.0 (0.96–4.11)	0.064	1.9 (0.91–4.00)	0.087
Educational status ^a^				
Educated (*n* = 108)	1		1	
Uneducated (*n* = 610)	1.5 (1.00–2.31)	0.047	0.9 (0.61–1.50)	0.858
Number of children				
1–3 (*n* = 377)	1		1	
Null (*n* = 19)	3.3 (1.14–9.25)	0.026	2.6 (0.91–7.64)	0.075
3–10 (*n* = 337)	1.2 (0.87–1.57)	0.286	0.7 (0.51–1.02)	0.066
Milk intake ^b^				
≥2 times a week (*n* = 273)	1		1	
<2 times a week (*n* = 282)	1.4 (0.98–1.93)	0.065	1.4 (0.96–1.92)	0.082
Instant noodle intake ^c^				
<2 times a week (*n* = 379)	1		1	
≥2 times a week (*n* = 226)	0.8 (0.61–1.17)	0.316	1.1 (0.78–1,55)	0.574
Smoking				
Never (*n* = 478)	1		1	
Former (*n* = 115)	1.4 (0.97–2.06)	0.071	1.1 (0.77–1.70)	0.498
Current (*n* = 140)	1.4 (0.96–2.17)	0.078	1.0 (0.68–1.61)	0.831
Alcohol intake				
Never (*n* = 510)	1		1	
Current (*n* = 94)	1.2 (0.81–1.77)	0.372	1.2 (0.79–1.76)	0.412
Former (*n* = 126)	0.8 (0.50–1.22)	0.279	0.8 (0.52–1.28)	0.385
BMI ^d^, kg/m^2^				
Normal (*n* = 231)	1		1	
Underweight (*n* = 41)	1.2 (0.61–2.31)	0.624	1.0 (0.53–2.11)	0.860
Overweight (*n* = 280)	0.9 (0.63–1.27)	0.549	0.9 (0.67–1.38)	0.852
Obese (*n* = 177)	0.7 (0.47–1.92)	0.071	0.8 (0.54–1.23)	0.340
Waist circumference ^e^, cm				
<80 (*n* = 402)	1		1	
≥80 (*n* = 326)	0.8 (0.57–1.03)	0.076	0.8 (0.56–1.20)	0.324
Hypertension				
No (*n* = 415)	1		1	
Yes (*n* = 318)	1.4 (1.04–1.88)	0.024	1.2 (0.88–1.67)	0.238
HbA1c ^f^, mmol/mol				
Normal (*n* = 539)	1		1	
Elevated (*n* = 181)	1.8 (1.23–2.51)	<0.001	1.5 (1.06–2.19)	0.23
Total cholesterol ^g^, mmol/L				
Normal (*n* = 542)	1		1	
Borderline high (*n* = 129)	2.4 (1.60–3.53)	<0.001	2.4 (1.58–3.58)	<0.001
High (*n* = 53)	2.4 (1.32–4.28)	0.004	1.9 (1.06–3.52)	0.032
LDL-C ^h^, mmol/L				
Normal (*n* = 365)	1		1	
Borderline high (*n* = 317)	1.9 (1.37–2.56)	<0.001	1.7 (1.26–2.38)	<0.001
High (*n* = 40)	1.8 (0.93–3.45)	0.083	1.5 (0.75–2.92)	0.252
HDL-C ^i^ mmol/L				
Normal (*n* = 258)	1		1	
Low (*n* = 88)	0.9 (0.57–1.51)	0.765	1.1 (1.14–1.89)	0.605
High (*n* = 378)	0.7 (0.48–2.08)	0.263	1.1 (0.78–1.50)	0.632
TG ^j^, mmol/L				
Normal (*n* = 521)	1		1	
Borderline high (*n* = 107)	1.7 (1.11–2.55)	0.014	1.7 (1.08–2.64)	0.02
High (*n* = 80)	3.0 (1.80–4.91)	<0.001	3.1 (1.84–5.35)	<0.001

Abbreviations: COR, crude odds ratio; AOR, adjusted odds ratio for age and BMI; BMI, body mass index; HbA1c, glycated hemoglobin; LDL-C, low density lipoprotein cholesterol; HDL-C, high-density lipoprotein cholesterol; TG, triglycerides. ^a^ missing *n* = 15; ^b^ missing *n* = 178; ^c^ missing *n* = 128; ^d^ missing *n* = 4; ^e^ missing *n* = 5; ^f^ missing *n* = 10; ^g^ missing *n* = 9; ^h^ missing *n* = 11; ^i^ missing *n* = 53; ^j^ missing *n* = 22.

## Data Availability

Not applicable.

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
