# Peer review of "Vitamin D Status among Women in a Rural District of Nepal: Determinants and Association with Metabolic Profile—A Population-Based Study"

_nutrients, 2022, doi:10.3390/nu14112309_

Round 1
Reviewer 1 Report
This cross-sectional study aims to assess the prevalence of Vitamin D Deficiency and the association of VDD and metabolic parameters among a female population in a rural district of Nepal.
In the recent years, vitamin D has generated a keen interest among clinicians, public health specialists, and researchers. The publication output, including research studies, related to vitamin D has trebled in the last decade. The prevalence of vitamin D deficiency is reportedly increasing, and assay of serum 25-hydroxy vitamin D [25(OH)D] is one of the most frequently ordered nutrition-related blood investigations.
It is apparent that much of the reported high prevalence of vitamin D deficiency/insufficiency in healthy populations of the world is artificial, created by unjustified high cutoff values of serum 25(OH)D. Most of the individuals diagnosed as vitamin D-deficient based on these criteria lack any marker of ill health, including bone mineral deficiency.
There are some aspects that need to be clarified.
- As mentioned above, there is much controversy about the real health risks of vitamin D deficiency in healthy patients. Have any adverse effects been demonstrated in the investigated population (e.g. presence of osteopneia or bone pain in adults, rickettsia in children, etc.)?
- How were the eating habits of these subjects investigated? Why were only a few foods taken into consideration?
- The discussion of the paper needs to be improved. The authors should evaluate other possible effects that influenced the results (e.g. body fat is decisive because it sequesters blood vitamin D). The authors should also evaluate the cost/benefit of strategies, including fortification of foods with vitamin D, that may be evaluated and implemented so as to bridge the gap between current and recommended intakes of vitamin D.
Minor:
1. Remove the last sentence "This section is not mandatory but can be added to the manuscript if the discussion is unusually long or complex."
2. There are small plagiarised parts between the methods (see attached file)

Author Response
Dear Reviewer 1
We would like to thank the reviewer for the thorough report and useful relevant comments and suggestions. We find that the comments have contributed considerably to the improvement of the manuscript.

Reviewer 2 Report
This is a cross-sectional study from rural Nepal focused on vitamin D status of women. The topic is of relevance and offers a unique opportunity to empower women in Nepal. As such the data needs to be reanalyzed in view of population health targets for vitamin D status. Details are outlined below.
Abstract
Vitamin D deficiency should be defined earlier. Having made this request, the terms do not match internationally accepted definitions. Deficiency is usually 25 to 30 nmol/L of serum 25(OH)D.
Line 28, is milk a source of vitamin D in Nepal? This is not clear in the abstract.
Lines 29-32, when obesity is added to the model does the pattern remain? Seems obesity would be a significant confounder.
The abstract needs to conclude in view of the cross-sectional data, pathogenesis is not feasible to examine in this design. The wording is far too strong.
Introduction
As above, appropriate definitions for vitamin D status need to be provided in order to place the rest of the introduction into context.
Line 51 suggests the present work is population oriented, the definition of vitamin D deficiency should thus be <25-30 nmol/L 25(OH)D.
Methods
The sample is very nice; the sampling time could be biased however with Oct-Dec sampling. This should be described in terms of ambient UVB at that time of year and clothing habits as well. Some additional information on how randomly selected and recruited would be of value.
Line 97, in place of “full blood” use “whole blood”.
Lines 78-81 are too brief.
Section 2.3, add the QC data for all of the analytes and certification programs participated in; show accuracy and precision of the QC samples, especially for serum 25(OH)D which requires international standardization – as the primary outcome. International standardization assays for other tests should also be outlined.
Line 136, n=80 missing data; imputation could overcome this, what type of data was missing? Add to a participant flow chart so the reader knows.
Line 150, milk intake needs to be described better in the methods, what type of milk, how much vitamin D is in that in Nepal and how was the survey conducted; assuming valid methodology. Frequency is reported, volume is unclear; as such is this a marker for other lifestyle or economic status?
Line 152, was dose of supplement available and if so what was it; was adherence examined?
Table 1; P values are shown but no post hoc test for groups/categories of 2 or more. Add the post hoc tests on the serum 25(OH)D data.
Table 2. The HDL looks very low, ensure this is commented upon. Table 2 could be more informative if the participants with 25(OH)D <30 or <50 nmol/L were compared to the opposite category for these biochemical variables. As I continued to read, this is essentially what is in table 4; thus table 2 is not needed. Table 4 requires post hoc tests.
Figure 1. this figure is not overly informative and is very large; the 95%CI are not included either. Delete it. It is essentially the same data as the first row of table 3.
Table 5, are all of these datasets normally distributed? Is the pattern linear? Some key figures should be shared to show the reader the type of correction pattern.
Line 209, use of “suffer” is not appropriate. Line 212 as well, avoid using such terminology.
Table 6, sample size for each row would be of help to the reader. The lower 95%CI in the COR for obese is likely an error (typo).
Discussion
Line 225, severe deficiency is introduced, <30 nmol/L is not severe in populations. Is the study powered to detect such a low prevalence? The 95%CI are not shown.
Line 232 also shows only 7% > 75 nmol/L. The data really should be reanalyzed with more appropriate categories such as <30, 30-50 and >=50 nmol/L to help with the sample size issues and the reference or comparator groups in the analyses.
Lines 250-254, clothing habits should also be discussed along with sun behavior relative to the rural work during the day. Is mid-day sun avoided and other indoor activity conducted then? How does clothing coverage change over the day of work if any? This is somewhat addressed lines 256-264, but avoidance of sun is not the same as avoidance of heat or work clothing needed.
UVB and UV-B are both used for the same term.
The positive association between BMI and vitamin D status is often not shown in other studies as too few are lean or underweight; there are some studies on this however that BMI <18.5 vs 18.5-24.9 are different. Cite some of those studies. The association is not linear.
The content of cows milk and synthesis by the cow is addressed in two different paragraphs and should be removed from one.
A serum 25(OH)D <30 nmol/L in 6.3% of the population is on par with that observed in many other countries. This does not come out in the paper and should.
Author Response
Dear Reviewer 2
We would like to thank the reviewer for the work, the thorough and up to date report, and the useful and relevant comments and suggestions. We believe that the reviewer's input has considerably contributed to improvement of the manuscript.
.

Round 2
Reviewer 1 Report
The authors responded to all my comments.